

# Evaluation and transcriptomic and metabolomic analysis of the ability of *Auricularia heimuer* to utilize crop straw

Di Zhang[1,2], Yuchen Liu[1], Ying Li[1], Guosheng Jiang[2], Mingzhu Meng[1] and Jihua Wang[1]

[1] School of Life Science and Technology, Harbin Normal University, Harbin, China
[2] Department of Smart Agriculture, Heilongjiang Agricultural Economy Vocational College, Mudanjiang, China

Corresponding author
Jihua Wang,
Wangjihua@hrbnu.edu.cn

## ABSTRACT

*Auricularia heimuer* is an important edible fungus, and the choice of its cultivation medium is very important to improve the yield and quality. Traditionally, *A. heimuer* mostly uses wood chips as cultivation substrate, but with the increase of agricultural waste, exploring agricultural straw as an alternative substrate has become a research hotspot. In this study, a wild *A. heimuer* strain W-ZD22 with good adaptability to straw matrix was used to measure mycelia growth characteristics and extracellular enzyme activity. Transcriptomics and non-targeted metabolomics methods were used to compare the effects of mycelia using agricultural straw matrix and wood chips matrix. It was found that the lignin enzyme activities of corn straw and wood chips were similar. By transcriptomic and metabolomic analysis, we further analyzed the transcription profiles of *A. heimuer* mycelia grown in different substrates (straw and corn stalk, straw and sawdust, corn stalk and sawdust), and identified 5,149, 2,740 and 2,933 different expression genes (DEGs), respectively. The three control groups had a total of 526 gene variants. The top 20 pathways with the highest concentration of DEGs mainly involved glyoxylate and dicarboxylate metabolism, glycine, serine and threonine metabolism, glycolysis/gluconeogenesis, pyruvate metabolism, oxidative phosphorylation, endoplasmic reticulum protein processing and ribosome. In order to further understand the similarity of enzyme activity of *Auricularia* mycelium on corn stalk and wood chips, metabolomic analysis of substrate of corn stalk and wood chips was conducted. It was found that different metabolites were significantly enriched in starch and sucrose metabolism, glutathione metabolism, carbon metabolism and other pathways, which provided theoretical basis for efficient comprehensive utilization of corn stalk in auricularia growth.

## INTRODUCTION

*Auricularia heimuer*, a very valuable genus of wood rot fungi with high nutritional value and efficacy (*Chen et al., 2008*), is the world's third most important cultivated mushroom (*Yuan et al., 2019*). It is widely planted in Asian countries such as China and Japan

(*Ren et al., 2022*; *Zhao et al., 2016*, *2019*) due to rapid industrial development brought about by the shortage and rapid rising costs of wood chips and other raw materials. Therefore, there has been a rise in research on alternative substrates for *A. heimuer* cultivation. Currently, it is feasible to use crop straw to partially replace wood chips in *A. heimuer* cultivation (*Huang et al., 2019*; *Ma et al., 2023*; *Ying et al., 2024*). However, there are still some complications in this process and accumulation may occur in large-scale *A. heimuer* production when using crop straw to completely or largely replace wood chips. Additionally, there is a lack of theoretical support.

According to the 2024 Global Straw Industry Research Report, the annual output of straw worldwide exceeds 2 billion tons at present, and mainly consists of straw from crops such as corn, wheat, and rice. The main carbon source of agricultural waste is lignocellulose, which is composed of three main polymers: cellulose, hemifiber, and lignin (*Guo, Wang & Lee, 2018*; *Zhou et al., 2023*). Due to its rich lignocellulose content, it is currently used as a potential substrate for producing edible fungi (*Dundar, Acay & Yildiz, 2009*; *Xu et al., 2020*). However, different fungi exhibit specific preferences for decomposing gene families depending on the substrate (*e.g.*, sawdust, straw). Wood decay fungi exhibit different lignocellulosic degradation abilities on different substrates, which greatly affects the growth rate, biomass production, and extracellular enzyme activity of mycelium due to the properties and composition of the lignocellulosic substrates (*Veloz Villavicencio et al., 2020*).

Cellulose, the main component of all plant materials and most abundant organic molecule on Earth, is a linear biopolymer of glucose molecules connected by β-1,4-glycosidic bonds. The enzymatic hydrolysis of cellulose requires a mixture of hydrolytic enzymes that act synergistically, including endoglucanases, exoglucanases (also known as cellulose disaccharide hydrolases), and β-glucosidases (*Dashtban et al., 2010*). Extracellular enzymes play an important role in the growth and development of edible fungi; therefore their activity has become an important measurement object (*Huang et al., 2019*). Edible fungi are usually cultured in a medium composed of sawdust, cottonseed husks, wheat bran, and other substances, which can induce the secretion of cellulose, hemicellulose, and other extracellular hydrolytic enzymes (*Huai-Liang, 2010*; *Lechner & Papinutti, 2006*; *Rani, Kalyani & Prathiba, 2008*). Lignocellulose lyase is a carbohydrate active enzyme that plays an important role in carbohydrate metabolism. The secretion of lignocellulose-degrading enzymes during the fermentation process of edible fungi is a necessary physiological function for their transformation during fungal growth (*Elisashvili et al., 2002*). Lignin is the most abundant renewable source of aromatic polymers on Earth, and its degradation is a necessary condition for carbon recovery (*Pollegioni, Tonin & Rosini, 2015*). Different substrate carbon sources induce different extracellular enzymes, which affect the growth and development of *A. heimuer*. However, research on cellulase, hemicellulase, and lignase from *A. heimuer* is still limited. Therefore, elucidating the secretion activity of these enzymes is valuable and may be of great significance for the effective application of substrate selection in crop straw cultivation during practical production. This study compared the activity of *A. heimuer* extracellular enzymes using various substrates and analyzed the differences in their ability to utilize straw substrates.

*A. heimuer* production can be produced from many lignocellulosic substrates, but sawdust is the most popular substrate in production. At the same time, there is a large amount of straw that needs to be treated every year in China's crops. Therefore, it is urgent to develop and utilize crop straw as a new substrate for growing *A. heimuer*, but little is known about the molecular mechanism of *A. heimuer* when using straw. We found that in the process of utilizing straw when cultivating *A. heimuer*, mushroom emergence experiments are often conducted with different ratios to guide production, but there have been few reports on the decomposition mechanism, especially regarding transcriptomics and metabolomics. Therefore, we used second-generation sequencing technology and LC-MS/MS technology to perform transcriptome sequencing and metabolite detection on *A. heimuer* mycelium cultured on a single different crop substrate, and detected differentially expressed genes (DEGs) and differentially accumulated metabolites (DAMs) involved in mycelial growth. This transcriptome information helped us understand the molecular mechanisms by which *A. heimuer* mycelium utilizes different crop straw. Our results can serve as an important basis for studying the mechanisms by which other wood decay fungi decompose crop straw.

## MATERIALS AND METHODS

### Characteristics and sources of *A. heimuer* strains

In this study, the fungus strain of *A. heimuer* (W-ZD22) was stored in the Germplasm Resource Bank of Heilongjiang Institute of Microbiology. The fruiting bodies of this strain were collected from willow trees near the Wolong River in Xichagou, Shiyan Town, Mudanjiang City (43°59′35.66″N, 129°27′58.15″E) in October 2020. After screening and purification, a strain with good tolerance to crop straw was obtained and named W-ZD22. Thirty varieties widely cultivated in the market were screened using pure straw culture medium with the temperature maintained at 25 °C and conventional conditions for ear emergence for *A. heimuer* cultivation. The shape of the ear piece is shown in Fig. 1.

### Crop straw treatment and element analysis

Various waste materials from crops, such as rice straws (D1), rice husks (D2), corn stalks (D3), corn cobs (D4), bean stalks (D5), and sawdust (D6), were collected. Crop residues and sawdust that were free from mold contamination were also collected, dried, ground finely, and passed through a 1 mm sieve to ensure the quality and reliability of the research materials. An elemental analyzer (EA3100) was used to measure TN and TC, and different substrates were weighed for three repeated values on the machine.

### Preparation of culture medium

In the solid culture medium formulations induced by different crop substrates, 200 g of crop straw was used as the main ingredient. The carbon nitrogen ratio was adjusted to a fixed value of 30:1 using a protein peptone with a nitrogen content of 14.5%. The main ingredients were soaked in four times their volume of water for 16 h, after which the juice was collected, boiled for 2 h, and then filtered. We added 18 g of agar up to a volume of 1,000 mL, divided 200 mL of the culture medium into 500 mL triangular flasks sterilized

| Dry fruit body | Ventral wrinkle | Reverse side wrinkle |
|:--:|:--:|:--:|
| 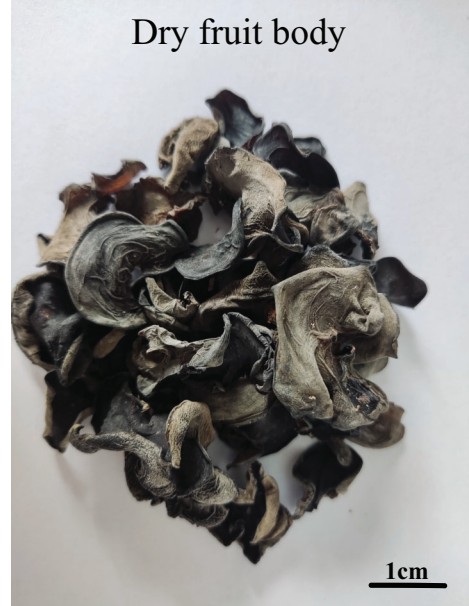 | 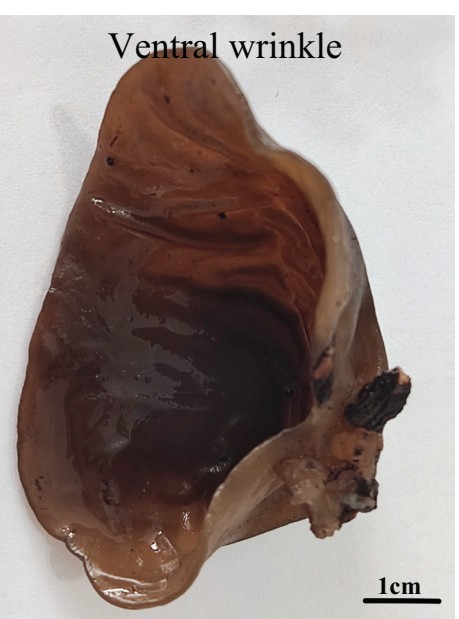 | 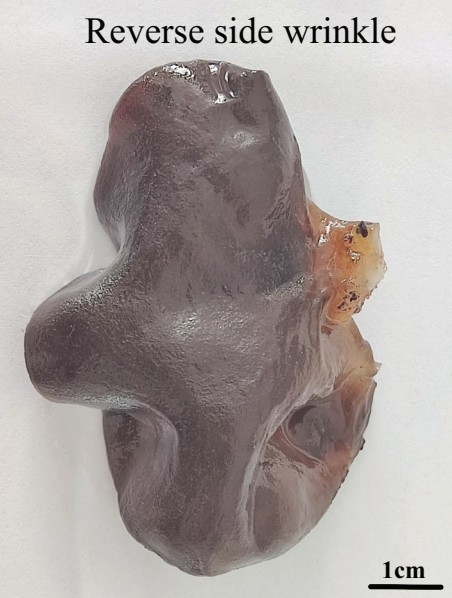 |

**Figure 1 Morphology of fruiting body of wild *Auricularia heimuer*.** Test materials.

with high-pressure steam at 121 °C for 30 min, and cooled naturally to around 60 °C. The medium was poured into a culture dish with a diameter of 9 cm on an ultra-clean workbench and left to solidify for later use. The liquid culture medium was prepared according to the proportion of solid culture medium formula (except for agar), and each formula was steamed to a constant volume of 1,000 mL. We divided 100 mL of liquid culture medium into 250 mL shake flasks, which we sterilized and cooled for later use.

## Observation of mycelial growth rate and colony morphology characteristics

Under sterile operating conditions, the tested strain was placed in the center of a sterile plate with different substrate culture media, inverted at 25 °C and cultured in a dark place. We observed and recorded the growth of hyphae, colony color, and uniformity of colony edges. The diameter of colony growth was measured and recorded using a vernier caliper cross method, and the growth rate (mm/d) was calculated. The experiment was repeated three times.

## Determination of cellulase, hemicellulase, and lignin enzyme activities

Extraction of crude enzyme solution: After 15 days of liquid culture, we divided the liquid culture into 50 mL centrifuge tubes, centrifuged them at 8,000 rpm for 10 min (4 °C), and the supernatant was taken as the crude enzyme solution to be tested. Filter paper assay (FPA) was performed using 50 mg filter paper strips (New China quantitative filter paper, approximately 1 cm × 6 cm). Endo-glucanase activity was determined using medical absorbent cotton. Exo-glucanase activity was determined using sodium carboxymethyl cellulose activity (Bodi Chemical Co., Ltd., Tianjin, China) (CMCase). β-glucosidase

activity was determined using a citrate buffer containing 0.5% salicin solution (D (-)-Salicin, Shanghai Maclin Biochemical Technology Co., Ltd.; citric acid, Damao Chemical Reagent Factory, Tianjin, China). Hemicellulase activity was determined using the xylan method (Maclin Reagents, Shanghai, China). Laccase (LACC) activity was determined using ABTS (Phygene Biotechnology, Fuzhou, China). The activity of peroxidase (Lip) was determined using sodium tartrate buffer and resveratrol (Maclin Reagents, Shanghai, China). The activity of manganese peroxidase (Mnp) was determined using $MnSO_4$ (Guo Yao Chemical Reagents, Beijing, China). The amount of β-glucosidase, CMCase, and cellulase that produces 1 μmol glucose per hour is defined as 1 U. The amount of hemicellulase that causes a 0.01 change in OD value is defined as 1 U. LACC and Lip activities are defined as 1 U when 1 μmol of substrate is catalyzed within 1 min. DNS is added to terminate the enzymatic reactions of β-glucosidase, CMCase, cellulase, hemicellulose, and amylase, and then boiled in a water bath. All enzymes were determined using spectrophotometry.

## Transcriptome data processing and bioinformatics analysis

Collection of mycelium samples: the mycelium at the bottom of 50 ml centrifuge tube was poured onto sterile filter paper, and mycelium balls were picked out with sterile tweezers and transferred to a 1.5 ml centrifuge tube. After quick freezing with liquid nitrogen, the mycelium was stored in an ultra-low temperature refrigerator at −80 °C, and transcriptome analyses were performed using three biological replicates per sample.

After RNA extraction, purification, and library construction, the samples were sequenced using next generation sequencing (NGS) technology. A power analysis for RNA-seq was performed using the R package RNASeqPower (version: v1.38.0) with parameters set to effect = 2 and alpha = 0.05. RNASeqPower was 0.853940898.

The raw data were filtered to obtain high-quality clean data, and Trinity software was used to assemble clean data into unigenes. The functional annotation, expression analysis, and structural analysis of unigenes were performed. The databases used for gene functional annotation included NCBI non-redundant protein sequences (NR), Gene Ontology (GO), Kyoto Encyclopedia of Genes and Genomes (KEGG), evolutionary genetics of genes: non-supervised orthologous groups (eggNOG), Swiss Prot, and Pfam. The clean reads of each sample were aligned to the reference sequence using the transcriptome expression quantitative software RSEM. The number of reads on each gene was then counted and the FPKM value of each gene was calculated for the following analysis. Differential gene expression analysis was performed using DESeq, and the screening conditions for DEGs were: expression fold |log2FoldChange| > 1, significant $p$-value < 0.05. GO enrichment analysis on DEGs was performed using topGO, and KEGG enrichment analysis on DEGs was performed using clusterProfiler.

## Non-targeted metabolomics analysis

### Sample preparation for metabolomics

An appropriate amount of the sample was mixed with pre-chilled methanol/acetonitrile/water solution (2:2:1, v/v). The mixture was vortexed, then subjected to cold

ultrasonication for 30 min. It was then incubated at −20 °C for 10 min, followed by centrifugation at 14,000 g for 20 min at 4 °C. The supernatant was collected and dried under vacuum. For mass spectrometry analysis, the residue was resuspended in 100 µL of acetonitrile/water solution (1:1, v/v), vortexed, and centrifuged at 14,000 g for 15 min at 4 °C. The supernatant was collected for instrumental analysis (*Doppler et al., 2016*).

### LC-MS/MS metabolomics experimental procedure

For the analysis of non-polar metabolites, an LC-MS/MS method was employed utilizing an UHPLC system (Vanquish; Thermo Fisher Scientific, Waltham, MA, USA) equipped with a Phenomenex Kinetex C18 column (2.1 mm × 50 mm, 2.6 µm) interfaced with an Orbitrap Exploris 120 mass spectrometer (Thermo Fisher Scientific, Waltham, MA, USA). The mobile phase consisted of two components: A (0.01% acetic acid in water) and B (isopropanol: acetonitrile, 1:1, v/v). The auto-sampler was maintained at 4 °C, with an injection volume of 2 µL. The Orbitrap Exploris 120 mass spectrometer was operated in information-dependent acquisition (IDA) mode, controlled by the Xcalibur acquisition software (Thermo Fisher Scientific, Waltham, MA, USA). In this mode, the software continuously assessed the full scan MS spectrum. The ESI source parameters were set as follows: sheath gas flow rate at 50 Arb, auxiliary gas flow rate at 15 Arb, capillary temperature at 320 °C, full MS resolution at 60,000, MS/MS resolution at 15,000, collision energy using stepped normalized collision energy (SNCE) at 20/30/40, and spray voltage at 3.8 kV (positive mode) or −3.4 kV (negative mode). The raw data were converted into the mzXML format using ProteoWizard and subsequently processed with a custom-developed program written in R, based on the XCMS package, for peak detection, extraction, alignment, and integration. The R package was applied in metabolite identification (*Han et al., 2024*). The identification of metabolites was based on the Human Metabolome Database (HMDB) (http://www.hmdb.ca), Metlin (http://metlin.scripps.edu), massbank (http://www.massbank.jp/), and mzclound (https://www.mzcloud.org) (*Cai et al., 2015*; *Zhou et al., 2022*).

### Statistical analysis

All data were measured three times and analyzed using the Statistical Package for the Social Sciences (IBM SPSS Statistics 21, IBM Corp., Armonk, NY, USA). Two-way analysis of variance and LSD test were used to compare the mean values of different groups, with a significance level of $p < 0.05$. The results were visualized using Origin 2021 software (OriginLab Corp, Northampton, MA, USA). Metabolic KEGG pathway enrichment, correlation analysis, cluster analysis, and Vip analysis using Scipy (Python) were performed. The correlation analysis employed the Pearson mathematical model.

## RESULTS

### Carbon and nitrogen content and C/N ratio of different substrates

During auricularia cultivation, nitrogen and carbon metabolism are the most important factors affecting mushroom growth. Carbon and nitrogen content were determined using an element analyzer and the results are shown in Table 1. The percentage of carbon

content in different substrates was between 37% and 50%. The carbon content in sawdust was the highest, reaching 47.74 ± 0.04. The lowest content of carbon in rice husks was 37.03 ± 0.14. Corn cobs (D4) and corn stalks (D3) had a closer carbon content to sawdust, making them more suitable as substrates for the growth of *A. heimuer* hyphae than rice straws and rice husks. In production, making crop straw into a more solid material should be considered in order to improve the carbon content and hardness. An appropriate C/N ratio can significantly promote the growth and yield of *A. heimuer* by facilitating its growth (*Geng et al., 2025*).

The nitrogen content in different substrates was below 1.7%, with the highest nitrogen content in soybean stalks reaching 1.637%. Second, corn stalks had a relatively high nitrogen content of 0.942%. The lowest content of nitrogen in sawdust was 0.381%.

In production, wheat bran is often used as the source of nitrogen nutrients with a nitrogen content of 1.553%. When cultivating *A. heimuer*, using bean stalks and corn stalks should be considered in order to supplement the nitrogen nutrients. The C/N rate is relatively large among different substrates, ranging from 24:1 to 125:1. It is generally recommended to adjust the C/N rate to 30:1 during the growth stage of mycelium. Overall, in terms of carbon and nitrogen content, the most suitable crop substrates for cultivating *A. heimuer* hyphae are rice straw (D1), corn stalk (D3), and bean straw (D5).

The ratio of cellulose, hemicellulose, and lignin varies with the source and type of lignocellulose (*Zhou et al., 2011*). Corn stalks had the highest cellulose content, 49.56 ± 0.34; corn cob had the highest hemicellulose content, 35.01 ± 1.12; and sawdust had the highest lignin content, 25.69 ± 0.50. From the perspective of lignin content, corn cobs and corn stalks are the best substitutes for sawdust.

## Effects of different substrates on mycelial growth

Among the five substrates, *A. heimuer* W-ZD22 had a better growth rate against corn stalk (D3, 76.88 ± 0.23), bean stalk (D5, 77.33 ± 0.19), and sawdust (D6, 76.63 ± 0.33), as shown in Table 2 and Fig. 2. There was no significant difference between D5, D3, and D6.

The main reason for the fastest growth rate of D5 was the high nitrogen content of pure straw in D5. This was consistent with high nitrogen content and fast growth rate of mycelium in production. However, although the mycelium growth rate of D5 was fast, the mycelium of D5 was relatively small. We also found that D4 had a good effect on cultivating *A. heimuer* mycelium, which was consistent with previous experimental results. According to mycelia density, color, and colony edge uniformity, D3, D4, and D6 had better adaptability and D6 was the best substrate.

We also found that D4 had a good effect on cultivating *A. heimuer* mycelium, which confirmed the potential of corn cob as a substrate for edible fungi cultivation (*Castorina et al., 2023*; *Fang et al., 2024*; *Geng et al., 2025*). In terms of mycelium density, colony color, and colony edge uniformity, D3 and D6 had the best adaptability, and D3 was selected as the best straw substrate in this article.

**Table 1 Composition of lignocellulose and C, N from different substrates.**

| Material | Cellulose (%) | Hemicellulose (%) | Lignin (%) | C (%) | N (%) | C/N (integer) |
|----------|---------------|-------------------|------------|-------|-------|---------------|
| D1 | 45.87 ± 0.24ab | 25.43 ± 0.72b | 15.63 ± 0.36c | 37.44 ± 0.05e | 0.79 ± 0.01c | 48:01:00 |
| D2 | 43.99 ± 0.17bc | 14.98 ± 0.72d | 14.83 ± 0.80c | 37.03 ± 0.14f | 0.54 ± 0.01e | 69:01:00 |
| D3 | 49.56 ± 0.34a | 23.77 ± 0.86b | 19.56 ± 0.65b | 44.77 ± 0.22c | 0.94 ± 0.02b | 48:01:00 |
| D4 | 44.63 ± 1.23bc | 35.01 ± 1.12a | 20.31 ± 0.82b | 45.72 ± 0.19b | 0.66 ± 0d | 68:01:00 |
| D5 | 42.77 ± 0.62c | 24.88 ± 0.66b | 14.36 ± 0.55c | 40.04 ± 0.08d | 1.64 ± 0.01a | 24:01:00 |
| D6 | 40.02 ± 0.43d | 19.89 ± 0.80c | 25.69 ± 0.50a | 47.74 ± 0.04a | 0.38 ± 0f | 125:01:00 |

Note:

D1 is rice straw, D2 is rice husk, D3 is corn stalk, D4 is corn cob, D5 is bean stalk, D6 is sawdust, the difference is significant ($p < 0.05$). C/N standard value (integer) set to 30:1.

Letters a, b, and c represent significant differences, while the same letter indicates no significant difference.

## Enzyme activity analysis

### Cellulase activity

In 1976, *Mandels, Andreotti & Roche (1976)* discovered cellulase. Cellulase is a multicomponent enzyme that consists of C1 enzymes (cut inside glucanase endo-1,4-β-D-glucanohydrolase, EC3.2.1.4), Cx enzymes (circumscribed glucanase), and β-glucosidase (β-D-glucosidase, EC3.2.1.21), and is usually represented by FPA. The function of C1 enzyme is to hydrolyze natural cellulose into amorphous cellulose, Cx enzyme further hydrolyzes amorphous cellulose into fiber oligosaccharides, and β-Glucosidase hydrolyzes fiber oligosaccharides into glucose. Through significant analysis of cellulase activity, the results showed that *A. heimuer* hyphae had different effects on cellulase activity of different crop substrates. FPA activity results are shown in Fig. 3. Figure 3A shows that rice stem (D1), corn stem (D3), and soybean stem (D5) had higher FPA enzyme activity, with D1 being the highest at 1.53 ± 0.04. The FPA enzyme activity of corn cob (D4) and rice husk (D2) was relatively low;, and the FPA enzyme activity of sawdust (D6) was the lowest at 0.06 ± 0.03. Figure 4B shows that D1 had the highest C1 enzyme activity, 6.63 ± 0.06. Figure 4C shows that D1, D3, and D6 had higher Cx enzyme activities: 1.17 ± 0.06, 1.07 ± 0.03, and 1.17 ± 0.07, respectively. Figure 4D shows that D1 and D3 had high β-G enzyme activities of 3.32 ± 0.12 and 3.17 ± 0.07, respectively. There was a positive correlation with cellulose content, mainly due to the higher cellulose content in rice husk and corn stalks. *A. heimuer* mycelium activity in relation to cellulase determines the decomposition ability of mycelium to substrates to a certain extent, so it is indicated that *A. heimuer* has the best utilization rate of D1 and D3.

Cultivation substrate plays a crucial role in the yield, mycelial growth rate, biological efficiency, and nutritional parameters of edible and medicinal mushrooms (*An, Wu & Dai, 2019*). The ability of heimuer to utilize corn cobs as a substrate and produce cellulose-degrading enzymes has been confirmed (*Yu et al., 2022*). Enzyme activity assays indicated that the filter paper enzyme activity of heimuer significantly increased when cultivated on a corn cob substrate. This is consistent with the high cellulase and filter paper enzyme activities of D1 and D3 in this study. Research has found that the activity of cellulase is closely related to the growth and yield of heimuer. Optimized extraction

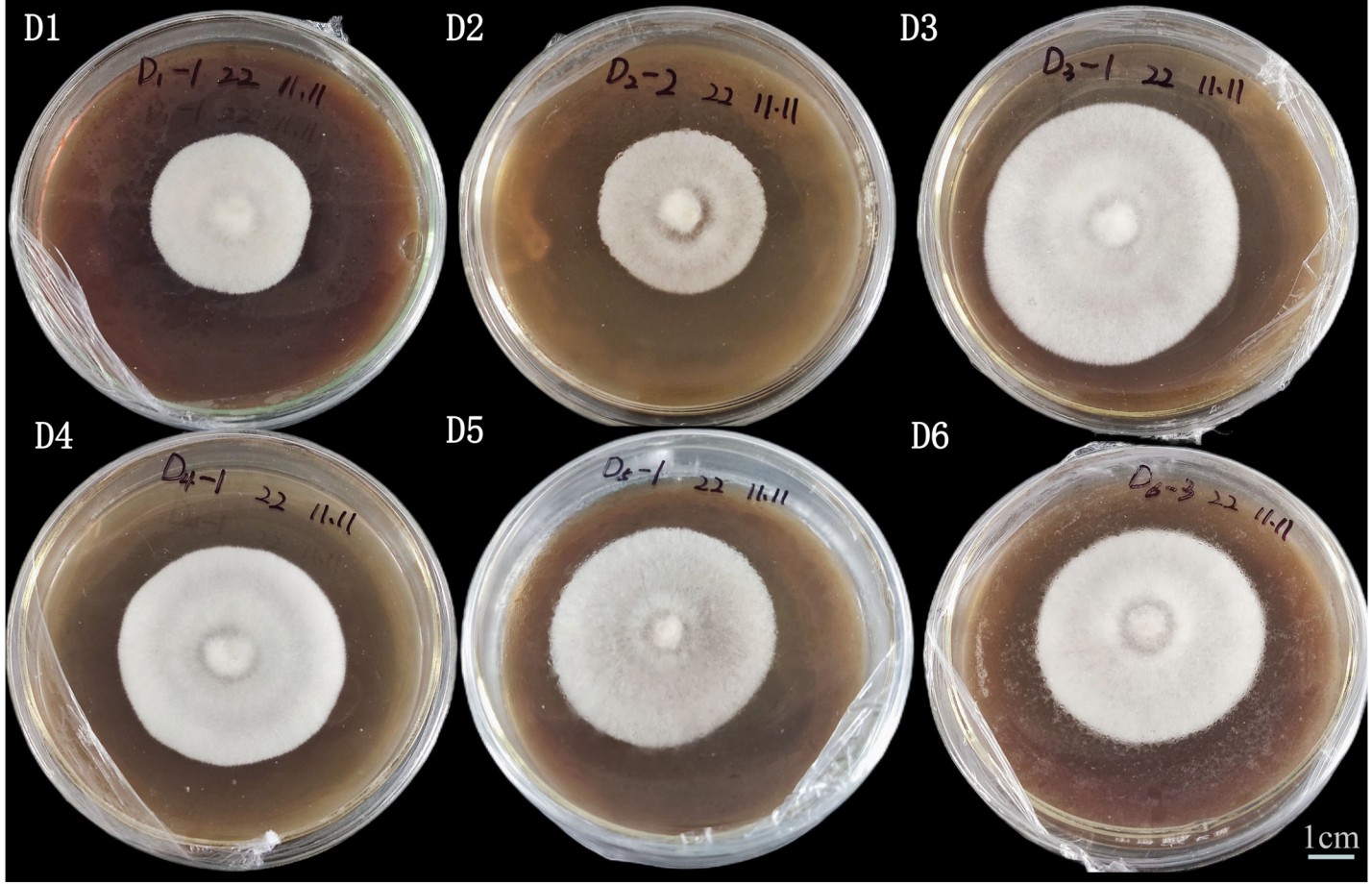

**Figure 2 Image of adaptability and colony morphology characteristics of mycelial straw on the 8th day.** Strain culture characteristics: 25 °C cultivation is optimal, the age is 10~12 days, neat colony edges, and strong mycelial vitality. D1 is rice straw, D2 is rice husk, D3 is corn stalk, D4 is corn cob, D5 is bean stalk, D6 is sawdust.

**Table 2 Effects of different substrates on mycelial growth.**

| Sample | Mycelial growth adaptability | | | | |
|---|---|---|---|---|---|
| | Mycelium density | Mycelium color | Edge of colony | Colony pigment | Mycelial growth rate (mm/d) |
| D1 | ++ | White | Neat | None | 44.00 ± 0.19d |
| D2 | ++ | Gray | Neat | None | 50.67 ± 0.00c |
| **D3** | **+++** | **White** | **Neat** | **None** | **76.88 ± 0.23a** |
| D4 | +++ | Gray | Neat | None | 71.17 ± 0.48b |
| D5 | + | Gray | Neat | None | 77.33 ± 0.19a |
| **D6** | **+++** | **White** | **Neat** | **None** | **76.63 ± 0.33a** |

Note:
"+" indicates sparse hyphae, "++" indicates sparse hyphae with average growth, and "+++" indicates dense and robust hyphae. The mycelial growth rate is the 12th day mycelial growth rate, with three replicates, and each replicate is measured three times to take the mean and standard error. D1 is rice straw, D2 is rice husk, D3 is corn stalk, D4 is corn cob, D5 is bean stalk, D6 is sawdust. Different lowercase letters indicate significant differences in the growth rate of hyphae cultured on different substrates ($p < 0.05$). The bold font indicates that the mycelium has good growth adaptability on this substrate.

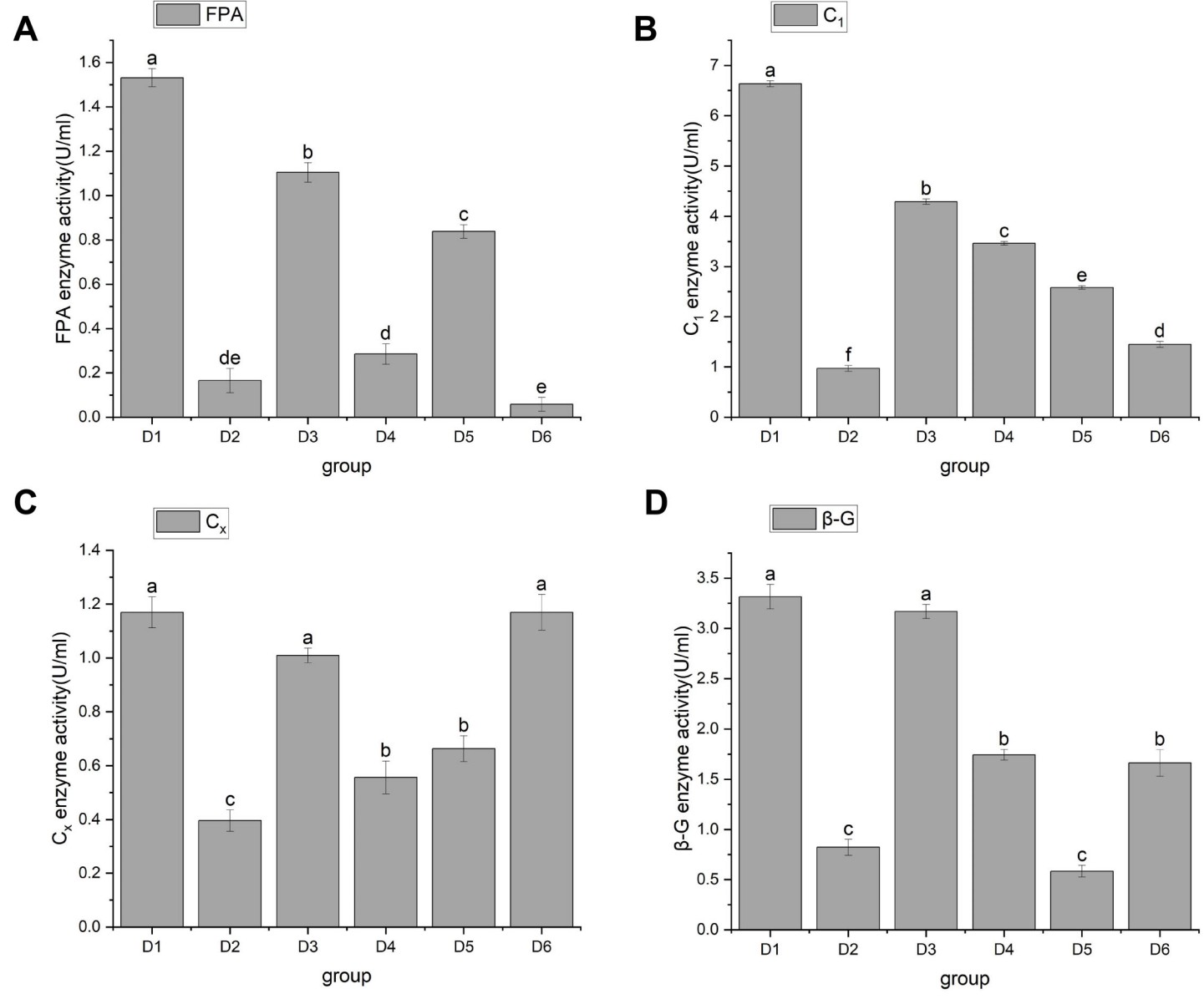

**Figure 3 (A–D) Changes in cellulase activity.** Note: FPA is total cellulase, $C_1$ is endoglucanase, $C_x$ is circumscribed glucanase, β-G is β-D-Glucosidase. D1 is rice straw, D2 is rice husk, D3 is corn stalk, D4 is corn cob, D5 is bean stalk, D6 is sawdust. Letters a, b, c, d and e indicate significant differences, while the same letters indicate no significant differences.

conditions can significantly enhance the activity of cellulase, thereby promoting the growth of heimuer. The potential to effectively degrade cellulose and hemicellulose in corn cobs supports the use of corn cobs as a sustainable substrate for heimuer cultivation (*Geng et al., 2025*). We speculate that rice straw and corn stalks are also excellent agricultural straw substrates for heimuer cultivation. Therefore, we chose this substrate for subsequent experiments.

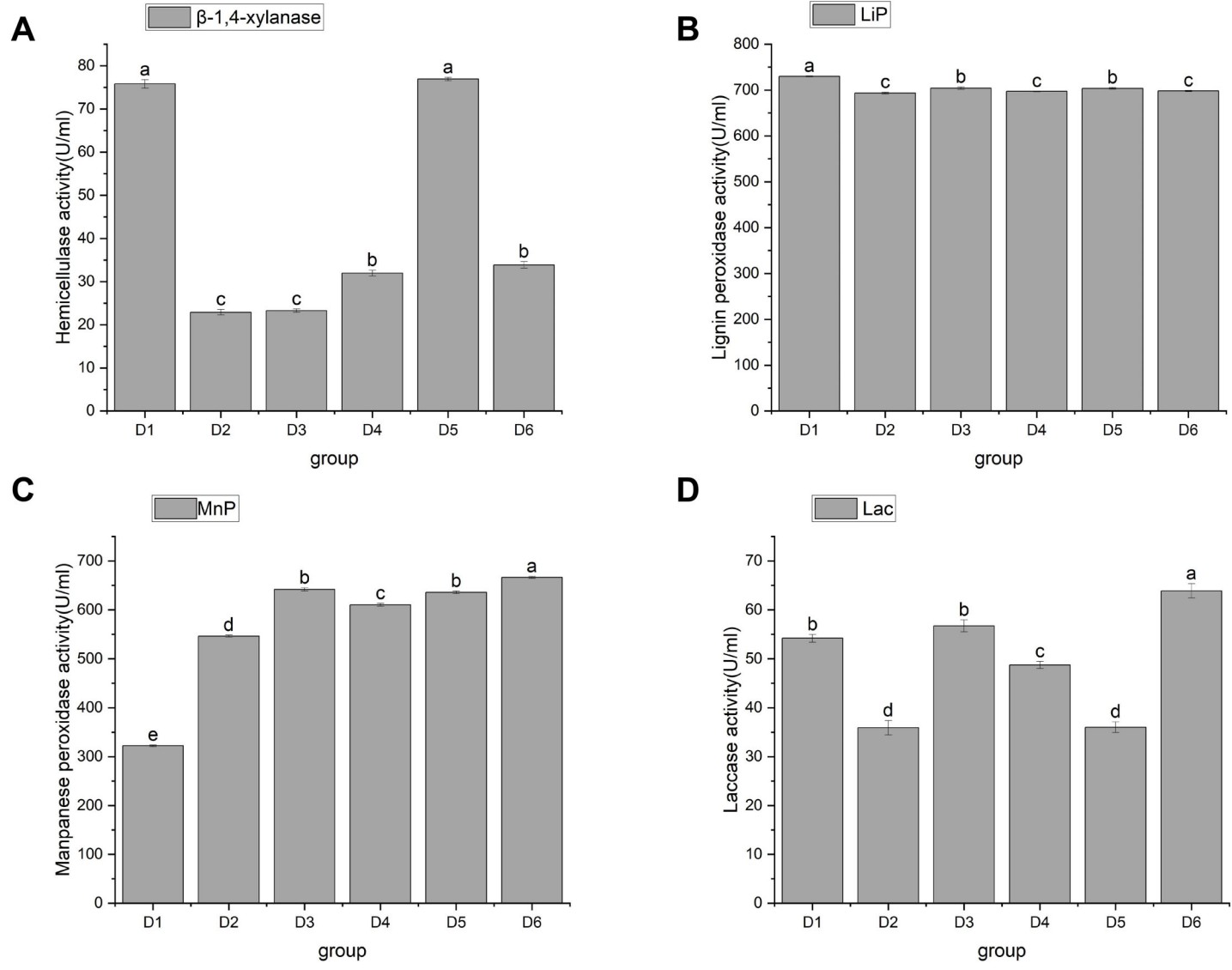

**Figure 4 Changes in activities of hemicellulase and ligninase.** Note: D1 is rice straw, D2 is rice husk, D3 is corn stalk, D4 is corn cob, D5 is bean stalk, D6 is sawdust. Letters a, b, c, d and e indicate significant differences, while the same letters indicate no significant differences.

### Hemicellulase and liginase activities

Hemicellulose is a large class of carbohydrates second only to starch and cellulose. For example, the content of hemicellulose in straw accounts for 25% to 50% of its dry weight (*Li et al., 2008*). The degradation of hemicellulose by edible bacteria is mainly caused by β-1, 4-endoxylanase, β-xylosidase, α-arabinofuranosidase, acetylxylanesterase, and α-glucuronidase. Endo-1, 4-β-xylanase (EC3.2.1.8) is abbreviated as xylanase. Xylanase, a key enzyme in the cellulase series, is mainly extracellular and acts synergically with xylosidase intracellular enzyme. Currently, xylanase has been the most extensively studied. As a physical barrier, lignin is embedded in cellulose and hemicellulose to form a dense

matrix, which hinders the biodegradation and utilization of straw (*Mei et al., 2020*). Currently, there are four enzyme activities that degrade lignin in plant cell walls (*de Gonzalo et al., 2016*; *Falade et al., 2017*): Lips (EC 1.11.1.14), MnPs (EC 1.11.1.13), and LACC/Lac (EC 1.10.3.2). These enzymes, as biocatalysts for lignin biodegradation, can increase the degradation capacity and speed. However, because lignin-decomposing enzymes are too large to penetrate undecayed plant cell walls, reactive oxygen species may be the factor that causes local lignin decay (*Pollegioni, Tonin & Rosini, 2015*). To sum up, it is necessary to investigate the correlation between the activities of hemicellulase and lignin enzyme and the growth of *A. heimuer* mycelia.

In this study, hemicellulase activity was highest in D1 and D5, reaching as high as 75.78 ± 1 and 76.92 ± 0.38 U/mL, respectively, with no significant difference between them. This was followed by D4 and D6, which were 31.96 ± 0.65 and 33.85 ± 0.76 U/mL, respectively, and then finally by D2 and D3, which had activity of 22.89 ± 0.65 U/mL and 23.27 ± 0.38 U/mL, respectively (Fig. 4A). D1, D3, and D6 showed significant differences in their ability to utilize hemicellulose. As shown in Figs. 4B, 4C, and 4D, agaric hyphae could produce Lac, MnP, and Lip simultaneously. Lip activity showed significant differences, with the highest being 729.66 ± 1a and the lowest being 693.14 ± 2.04c. Differential effects of fungi on single lignin-modifying enzyme (LME) production at different lignin growth substrates and nitrogen sources have been identified (*Rusitashvili, Kobakhidze & Elisashvili, 2024*). The activities of MnP and Lac in D6 were the highest, at 666.35 ± 2.03 and 63.89 ± 1.46, respectively, followed by D3, at 641.71 ± 3.22 and 56.71 ± 1.23, respectively. We speculate that the similarity in the utilization of LACC is related to the decomposition of dense lignin in corn stalk. Currently, only certain strains of WRF can simultaneously co-produce LACC, Lip, and MnP, making them highly valuable for research (*Wang, Yao & Su, 2018*).

### *Correlation analysis between different substrates, extracellular enzyme activity, and mycelial growth rate*

The C content of substrates mainly comes from lignocellulose. The content of cellulose, hemicellulose, and lignin varies in different substrates, which affects the ability of edible fungi to decompose substrates. The above experiment shows that different substrates, including carbon and nitrogen content, wood fiber content, and extracellular enzyme activity, have significant effects on the growth of *A. heimuer* hyphae. Consequently, we performed correlation analysis (Fig. 5) to analyze the potential relationship between carbon and nitrogen content, wood fiber composition, key extracellular enzyme activity in substrates, and the growth of *A. heimuer* hyphae. The results showed that there was a significant positive correlation between carbon content and lignin content in substrates. Furthermore, we also found that cellulose, hemicellulose, and lignin did not show a significant positive correlation with the growth rate of W-ZD22 mycelium ($0.05 \geq p > 0$), and the difference was not significant.

In addition, LACC also showed its multiple effects in this study. It not only has a significant positive correlation with lignin content, but also significantly affects the activity of exoglucanase, further highlighting its complex role and importance in lignocellulosic degradation.

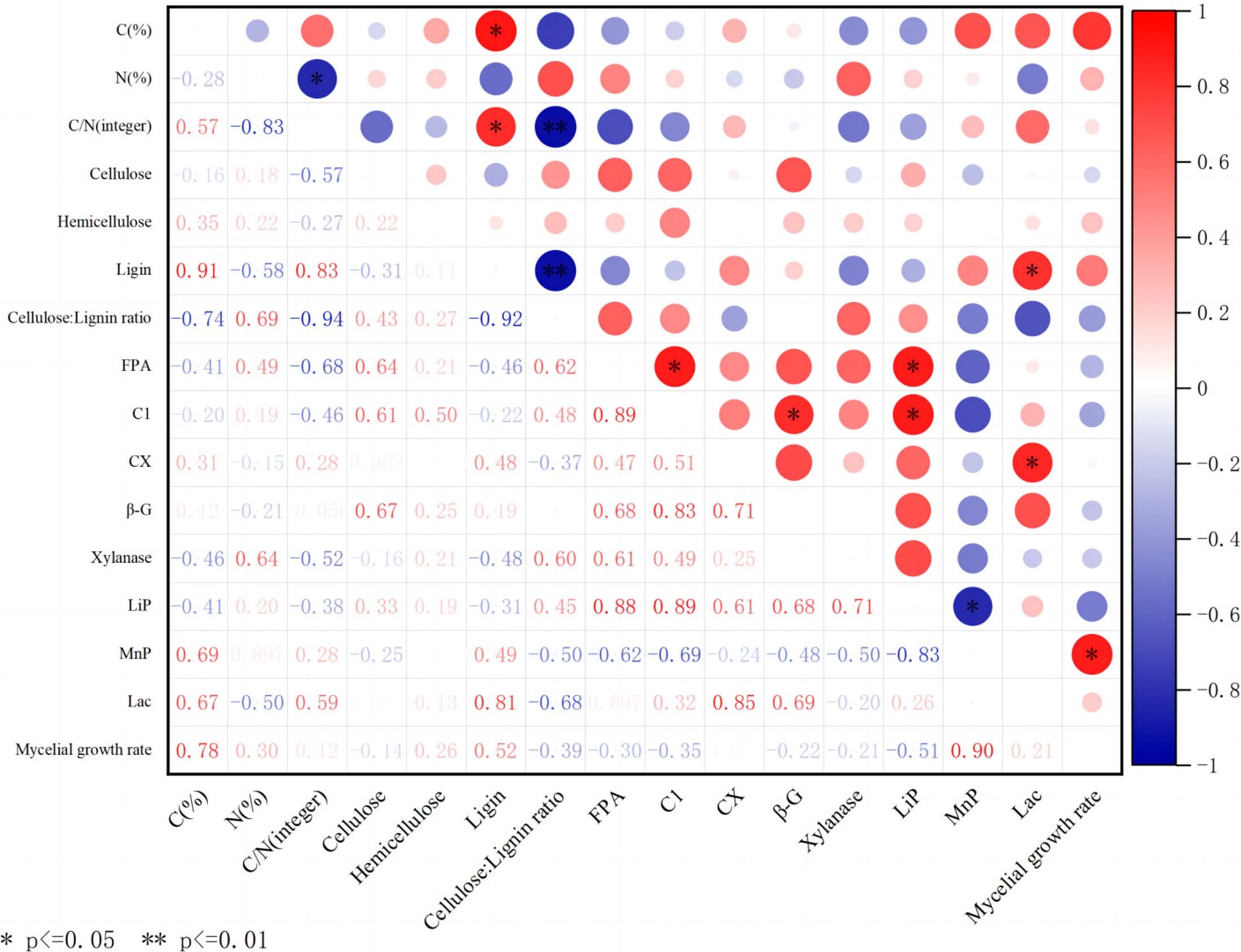

* p<=0.05    ** p<=0.01

**Figure 5 Correlation heatmap between different substrates, extracellular enzyme activity, and mycelial growth rate.** Red indicates a positive correlation; green indicates negative correlation. The shade of color indicates the degree of correlation. The asterisk (*) represents significant correlation. C(%) represents carbon content; N(%) represents nitrogen content; C/N is the carbon-to-nitrogen ratio; Cellulose is the content of cellulose; Hemicellulose is the content of hemicellulose; Ligin is the content of lignin; Cellulose/Lignin ratio is the ratio of cellulose to lignin content; FPA is the total cellulase activity; C1 is the endo-glucanase activity; CX is the exo-glucanase activity; β-G is the β-glucosidase activity; Xylanase is the xylanase activity; Lip is the peroxidase activity; MnP is the manganese peroxidase activity; Lac is the laccase activity; Mycelial growth rate is the growth rate of mycelium.

Based on the above findings, we selected representative substrates with decomposition ability (D1, D3, D6) for transcriptome and metabolomics studies. These studies will further analyze the differences in transcription and metabolism of *A. heimuer* in the process of using crop stalk and other substrates, particularly in the expression and regulation of key enzymes such as endoglucanase, exoglucanase, MnP, and LACC.

## Transcriptomic analysis

### DEG analysis

We used RNA-seq to detect DEGs on D1, D3, and D6 substrates during the growth of
*A. heimuer* hyphae. 5,149, 2,740, and 2,933 DEGs were detected in the D1-*vs*-D3, D1-*vs*-
D6, and D3-*vs*-D6 groups, respectively. For the comparison group of D1-*vs*-D3, 2,689
DEGs were upregulated and 2,460 DEGs were downregulated. For the comparison group
of D1-*vs*-D6, 1,296 DEGs were upregulated and 1,444 DEGs were downregulated. Finally,
in the comparison group of D3-*vs*-D6, 1,599 upregulated DEGs and 1,374 downregulated
DEGs were obtained (Fig. 6A). As shown in Fig. 6B, 526 consensus DEGs were prevalent in
the three comparison groups, which indicates that the expression of these genes are
consistent in *A. heimuer* hyphae in response to different substrates. Across the three
comparison groups, there were 507 (D1-*vs*-D6), 1,499 (D1-*vs*-D3), and 470 (D6-*vs*-D3)
specific DEGs. The D6-*vs*-D3 comparison group had the least specific DEGs, which we will
focus on later.

To understand the expression patterns of DEGs on the three substrates, bidirectional
clustering analysis was performed on the union and samples of the differential genes of all
comparison groups (Fig. 6C). The heatmap showed that the biological duplication of the
samples was good. The three groups of DEGs were mainly divided into highly-expressed
genes (red) and low-expressed genes (green). These 526 consensus DEGs showed the
expression levels of the same gene in different samples and expression patterns of different
genes in the same sample, with the expression patterns of D1 and D6 being most similar.

### Functional annotation and enrichment analysis of DEGs

In order to study the biological processes involved in these DEGs in three comparative
groups, GO annotation was performed on the DEGs present in each comparative group.
GO classification was performed based on molecular function (MF), biological process
(BP), and cellular component (CC). The GO annotations were mapped to GO terms, and
the number of annotated genes on the second level classification are shown in Fig. 7A.
These DEGs are significantly enriched in the metabolic processes of BP, the cell and
organelle parts of CC, and binding and catalytic activity of MF.

KO annotation was completed by KOBAS annotation system and statistical results of
the KEGG pathway were obtained (Fig. 7B). All 35 second level branches under five levels
(metabolism, genetic information processing, environmental information processing,
cellular processes, organic systems) have mappings. The dominant pathways are
carbohydrate metabolism, signal transduction, transport, and catabolism. Lignocellulase
genes mainly exist in these metabolic pathways, which are also the key pathways to pay
attention to.

To further analyze DEG-related pathways, these transcripts were KEGG annotated and
enriched to 111, 106, and 102 pathways in the three comparison groups, respectively.
Based on the previous experiments, we focused on the analysis of the D6-*vs*-D3 group. All
single-feature genes were functionally predicted and classified in KEGG database. They
were divided into five main functional categories and 18 sub-categories (Fig. 7B). The
largest category was the metabolic cluster, which accounted for about 69 items, and of

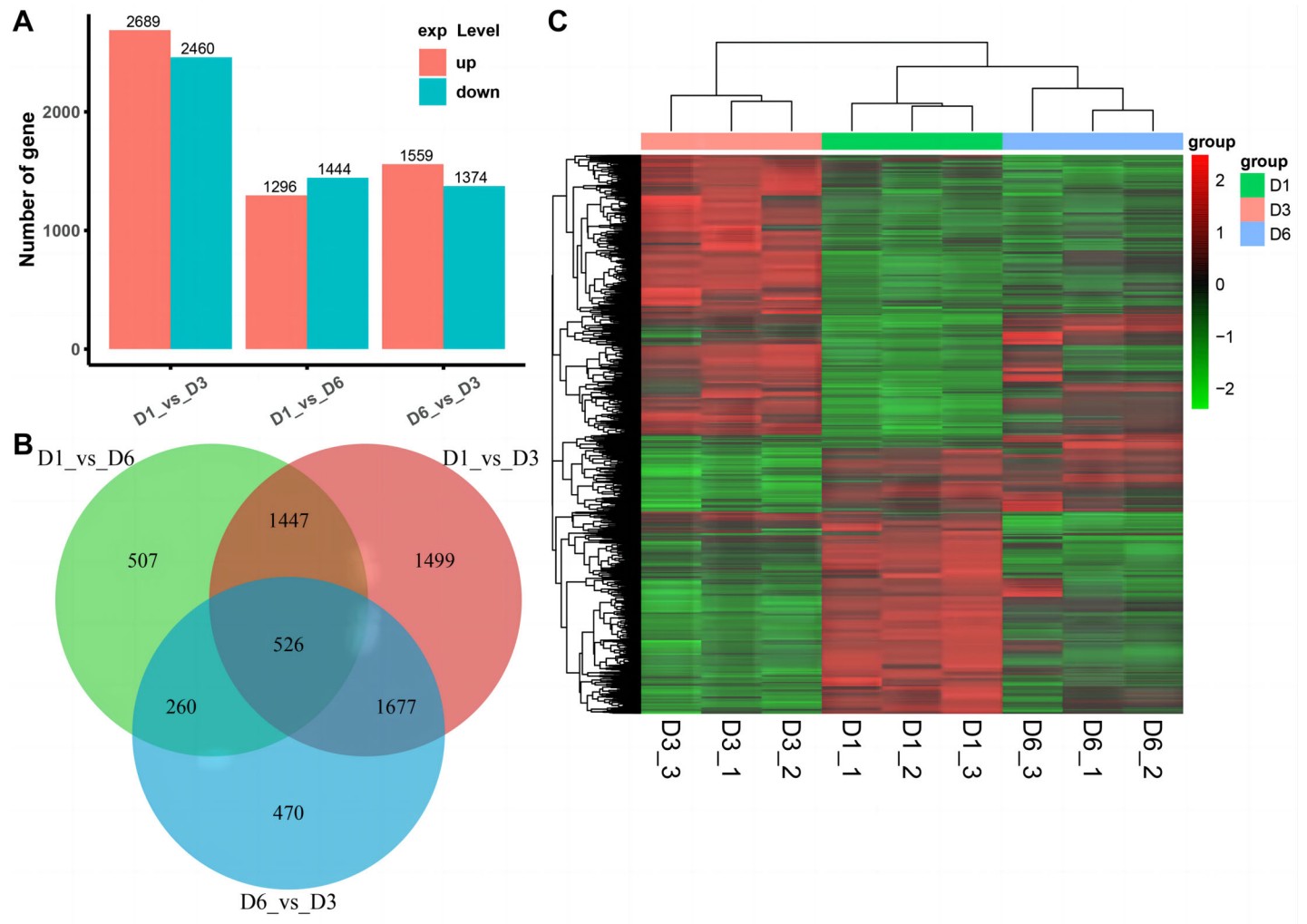

**Figure 6 Analyzes the differentially expressed genes (DEG) occurring in the three comparison groups.** (A) Stacked bar chart of the number of DEGs in different comparisons. (B) The Venn diagram of overlapping DEG in the three comparison groups. (C) Hierarchical clustering heatmap of 526 DEGs common in the three comparison groups. Red represents high-expressed genes and green represents low-expressed genes

which carbohydrate metabolism (14 items, 13.7%) was in the subcategory. Figure 7C shows the bubble diagram of 14 enriched carbohydrate metabolism pathways, and 15.66% DEGs were enriched in the starch and sucrose metabolism pathways (ko00500).

## Metabolomics analysis

### Screening and analysis of differential metabolites

In order to better understand the metabolic mechanisms of different substrate media during the growth process of *A. heimuer* hyphae, non-targeted metabolomics analysis was conducted on *A. heimuer* hyphae grown on different substrate formulation media. Under different carbon-induced conditions, the degradation ability of *A. heimuer* strains to lignin, cellulose, and hemicellulose varies. LC-MS analysis was performed on the samples treated with fermentation culture. As shown in Fig. 8A, PCA results showed good sample

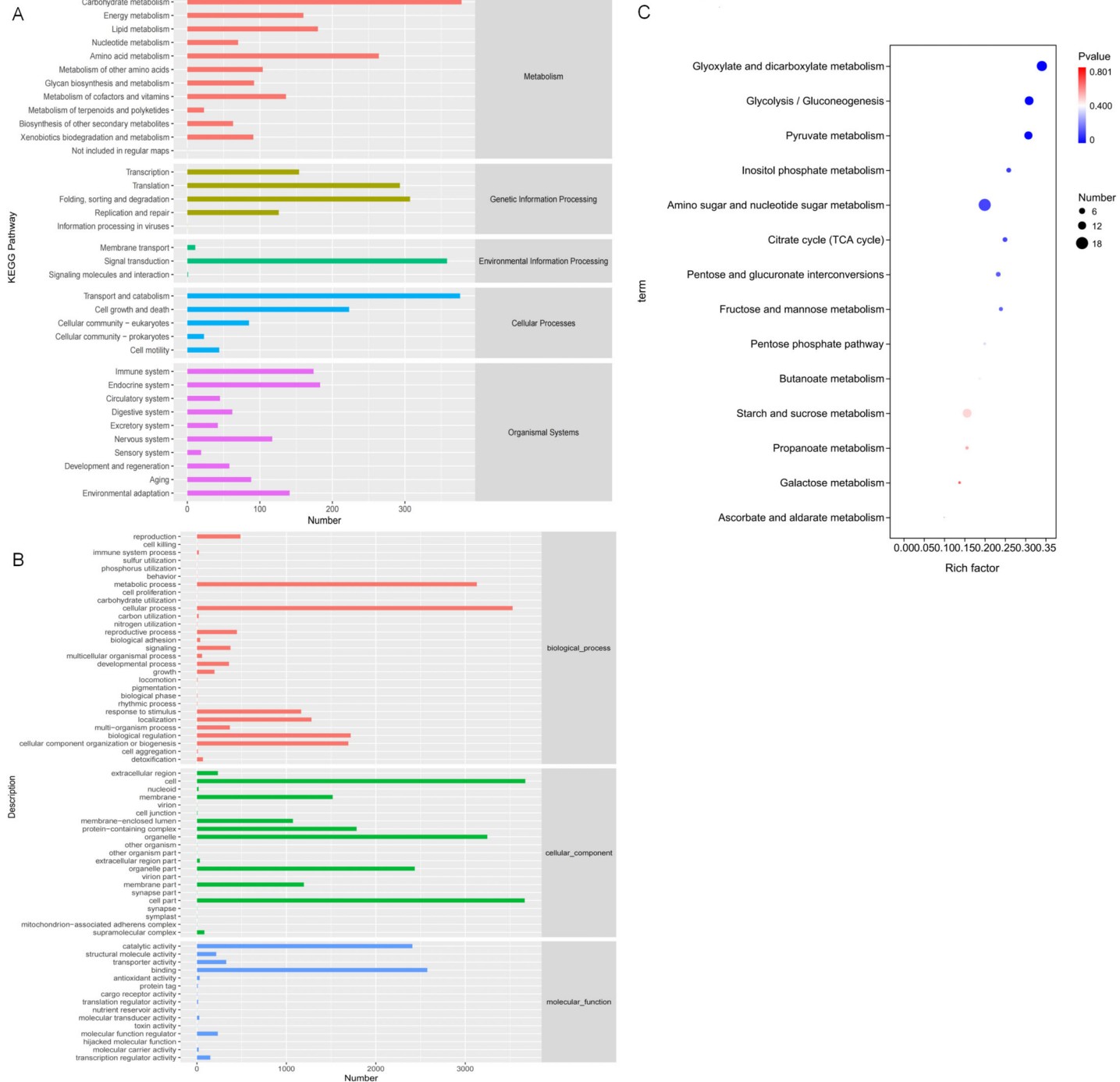

**Figure 7 Gene Ontology (GO) and KEGG annotations of DEG.** (A) The GO term with the highest degree of DEG enrichment among the three comparison groups; (B) the KEGG Pathway with the highest degree of DEG enrichment among the three comparison groups; (C) bubble diagram of carbohydrate metabolism pathways.

repeatability, indicating that metabolome data could be used for subsequent analysis. For the comparison of inter-group differences between different groups, the supervised orthogonal partial least squares discriminant analysis (OPLS-DA) model was used, and

permutation tests were used to identify that supervised learning methods did not obtain classification by chance (Fig. 8B). Using this approach, it is possible to better distinguish between group differences and improve their effectiveness and parsing ability (*Meng et al., 2022*).

Non-targeted metabolomics analysis showed that 453 differential metabolites of D1-*vs*-D6 were up-regulated and 422 were down-regulated, 282 differential metabolites of D6-*vs*-D3 were up-regulated and 233 down-regulated, and 362 differential metabolites of D1-*vs*-D3 were up-regulated and 343 down-regulated (Fig. 9A). The differential metabolites were clustered (Fig. 9B). The top 20 metabolites included (Fig. 9C) phloroglucinaldehyde, (4-(β-D-Glucopyranosyloxy) phenylacetic acid, (R)-2-hydroxystearate, luminol, gentisic acid, 3,4-dihydro-6-hydroxy-alpha,2,5,7,8-pentamethyl-2H-1-benzopyran-2-pentanoic acid, 3alpha-hydroxyoreadone, fumaritine N-oxide, vicenin2, 1-(3,5-dihydroxyphenyl)-12-hydroxytridecan-2-one, 4-[2-[(1R,4aS,5R,6R,8aS)-6-hydroxy-5-(hydroxymethyl)-5,8a-dimethyl-2-methylidene-3,4,4a,6,7,8-hexahydro-1H-naphthalen-1-yl]acetyl]-2H-furan-5-one, 1'H-spiro[cyclopentane-1,2'-quinazolin]-4'(3'H)-one, pyridine, 2-hydrazinyl-5-nitro-3,4-dihydroxybenzoic acid, 11-deoxyprostaglandin E2, isoschaftoside, schaftoside, 2,3-dihydroxybenzoic acid, 4-oxododecanedioic acid, and armillyl_everninate.

### KEGG enrichment of differential metabolites

According to mycelial growth characteristics and enzyme activity, the utilization of corn stalk and sawdust was more similar. Therefore, D3 and D6 were selected for enrichment of metabolic pathways. The top 20 enrichment pathways were ABC transporters, metabolic pathways, biosynthesis of amino acids, arginine and proline metabolism, D-amino acid metabolism, C5-branched dibasic acid metabolism, pyrimidine metabolism, glyoxylate and dicarboxylate metabolism, 2-oxocarboxylic acid metabolism, caffeine metabolism, arginine biosynthesis, valine, leucine and isoleucine biosynthesis, arachidonic acid metabolism, β-alanine metabolism, biosynthesis of cofactors, starch and sucrose metabolism, glutathione metabolism, carbon metabolism, tyrosine metabolism, and phenylalanine metabolism (Fig. 10).

ABC transporters play important roles in various cellular processes (*Bonifer & Glaubitz, 2021*). In naphthylacetic acid-promoted cordycepin studies, significant enrichment of the ABC transporter pathway was found. ABC transporters are known to transport multiple amino acids, such as L-glutamate, and are involved in amino acid metabolism that affects cordyceps synthesis (*Wang et al., 2023*). Compared with classical amino acids, the distribution, metabolism and function of natural non-classical amino acids are still relatively obscure. Natural non-classical amino acids are mainly found in plants as secondary metabolites and play a variety of physiological functions. It is necessary to study and clarify the metabolic pathways and key enzymes of non-classical amino acids (*Zou et al., 2018*). An extracellular effector depends on the production and release of extracellular D-amino acids that regulate various cellular processes, such as cell wall biogenesis, biofilm integrity, and spore germination. Non-classical D-amino acids are mainly produced by broad-spectrum racemases (Bsr). The promiscuous nature of Bsr allows it to produce high concentrations of D-amino acids in environments with different

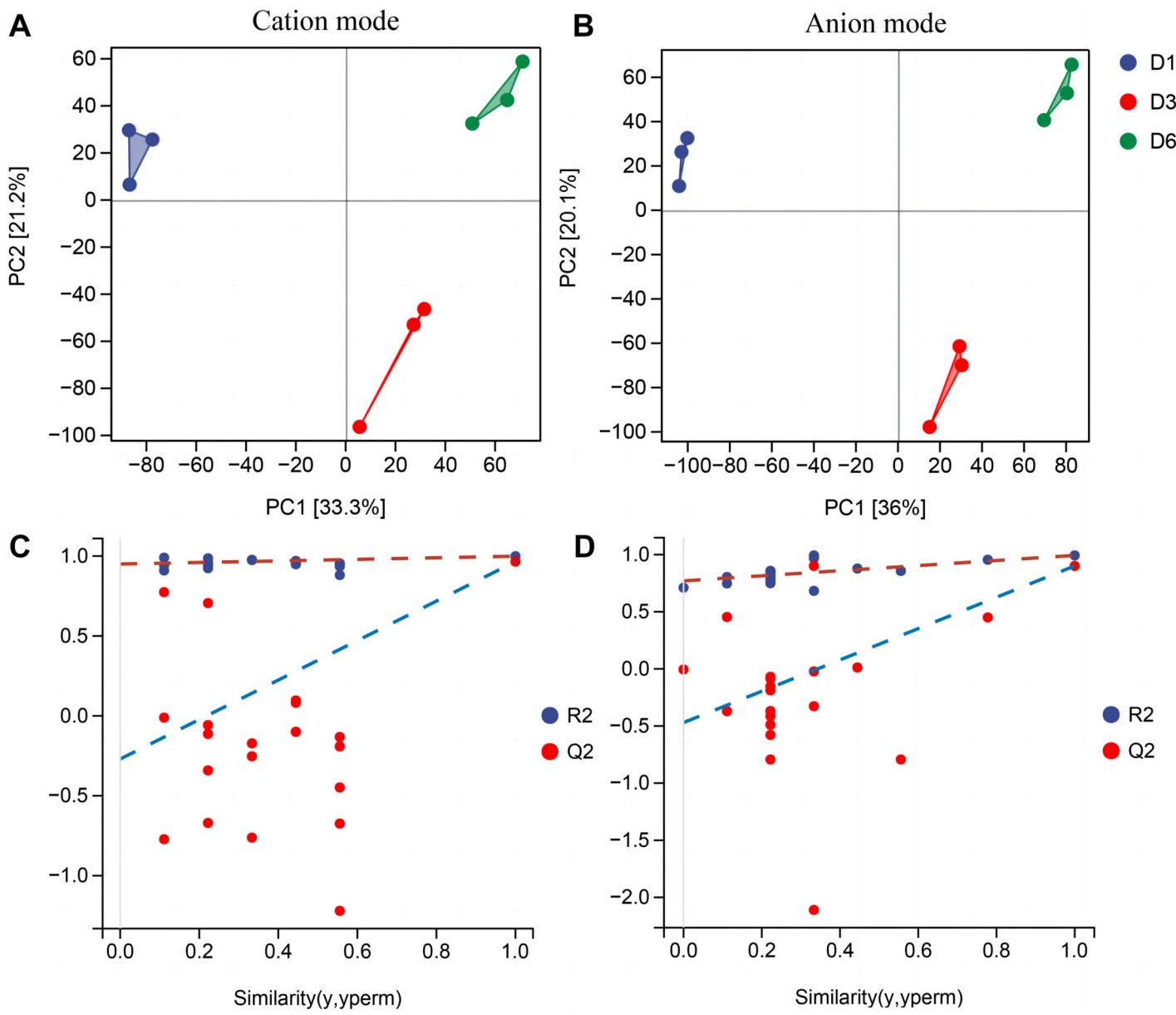

**Figure 8** **PCA and OPLS-DA analysis of non-targeted metabolomics.** Samples PCA scores of non-targeted metabolomics under positive ion mode (A) and negative ion mode (B); (B) OPLS-DA model establishment and substitution test results.

L-amino acid compositions. However, it has not been clear until recently whether these molecules exhibit different functions (*Aliashkevich, Alvarez & Cava, 2018*). Studies have shown that in *Candida albicans*, the TCA cycle not only occupies the central position of cell metabolism, but also regulates other biological processes such as $CO_2$ sensing and mycelial development through integration with the Ras1-cAMP signaling pathway (*Tao et al., 2017*). Over the past few decades, carbon catabolic inhibition (CCR) has fascinated scientists and researchers worldwide. This important mechanism allows priority to be given to energy efficient and readily available carbon sources over those that are less readily

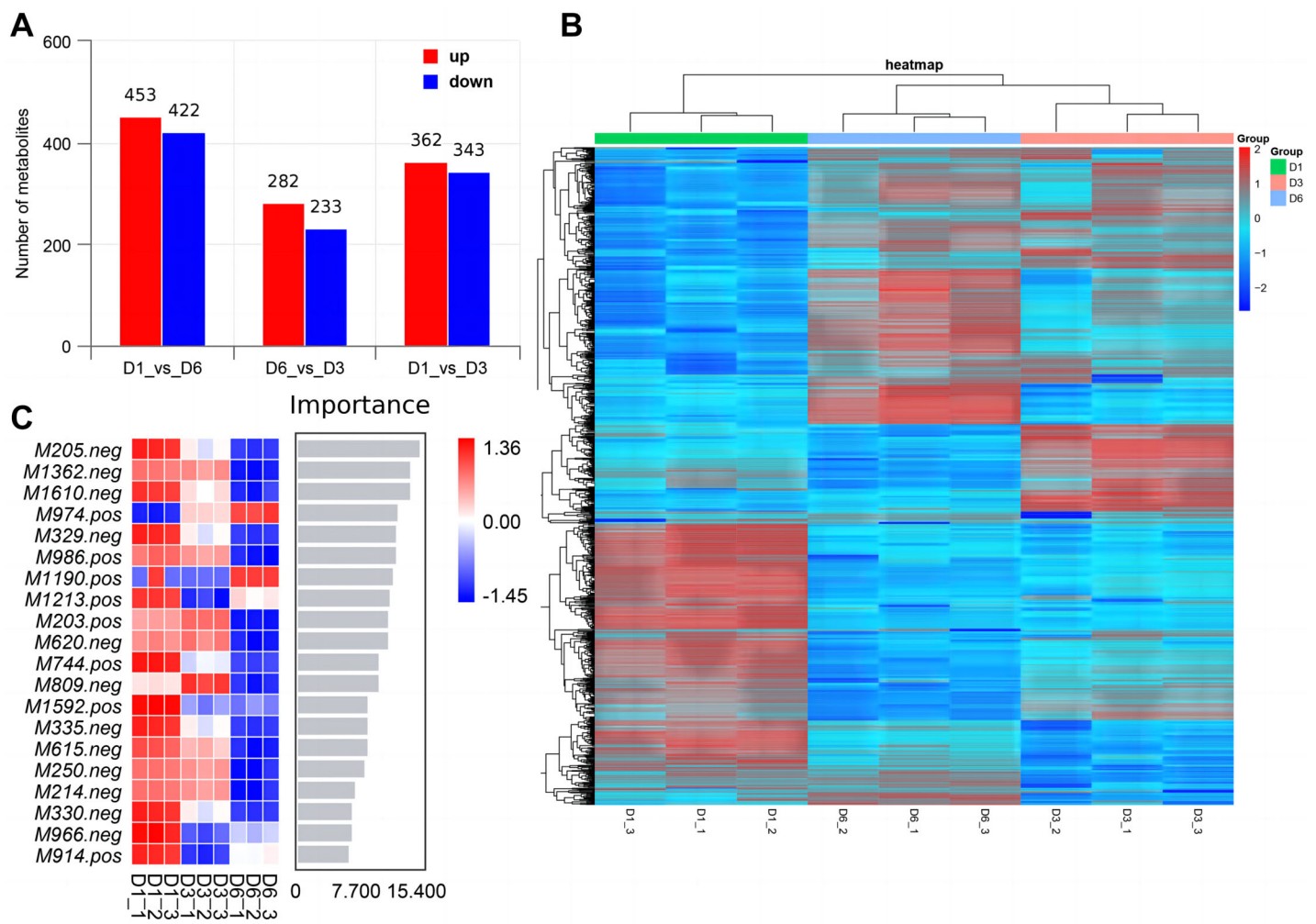

**Figure 9 Screening of differential metabolites.** (A) Histogram of differential metabolism in MERGE mode; (B) heatmap of differential accumulation metabolites; (C) top 20 differential metabolites.

available. This mechanism helps the microbes get the maximum amount of glucose to keep up with their metabolism. Microbes absorb glucose and highly favorable sugars before switching to less desirable carbon sources, such as organic acids and alcohols. In the CCR of filamentous fungi, CreA, as a transcription factor, is regulated to a certain extent by ubiquitination (*Adnan et al., 2017*).

Metabolic analysis showed that specific metabolites, including various amino acids, pyrimidines, acids, lipids, and carbohydrates, were produced when different substrates were decomposed. The synthesis and metabolism of amino acids have significant influences on the growth and metabolic product synthesis of fungi (*Erdmann et al., 2024*; *Verwaal et al., 2007*). Pyrimidine metabolism plays an important regulatory role in the growth and development of fungi (*Sha et al., 2025*), and acidic substances have significant effects on the flavor and nutritional components of *A. heimuer*. Carbohydrates are the

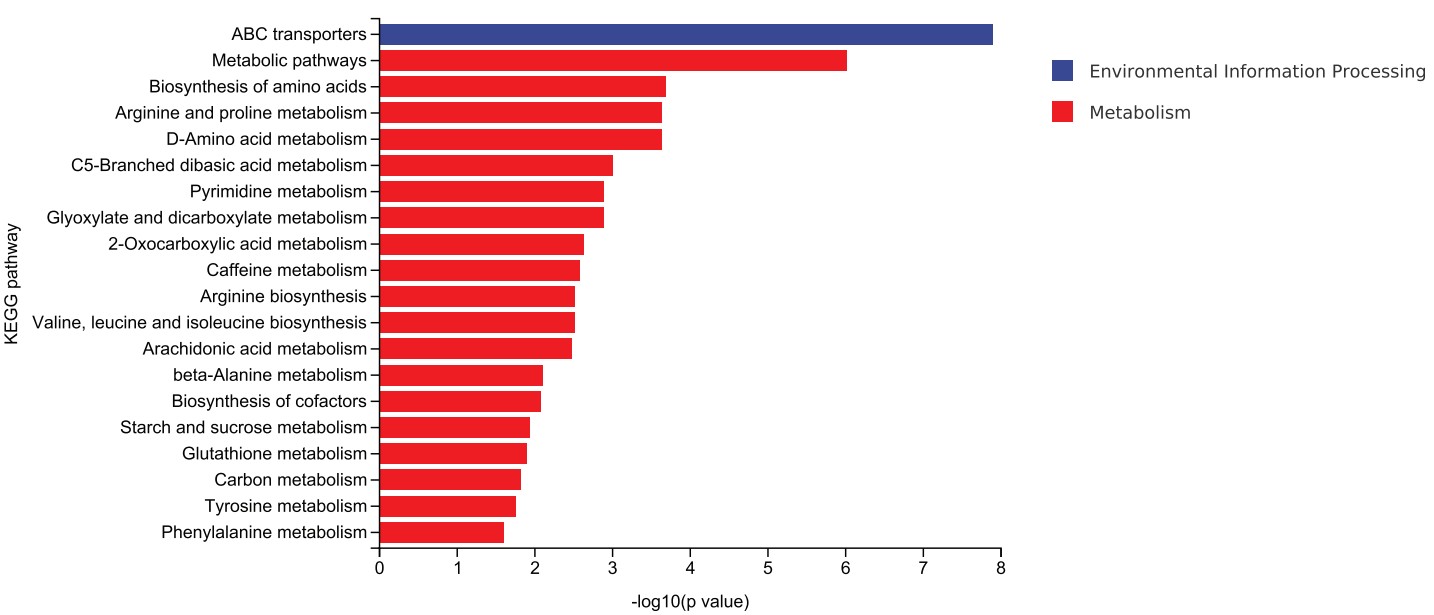

**Figure 10 Enrichment of the top 20 KEGG differential metabolic pathways in D3 vs D6.**

main energy source for *A. heimuer*, providing energy for growth and metabolism (*Yao et al., 2019*). The enrichment analysis showed that these substances had strong biological activity to promote catabolism and biosynthesis, and were involved in amino acid and sugar metabolism. Therefore, these metabolic pathways may be an important metabolic pathway in the production of *A. heimuer* from corn stalk. Studying the transcription factors that correspond to these metabolites facilitates targeted breeding. In particular, we focus on starch and sucrose metabolism, carbon metabolism, and redox reactions.

## Integrated analysis of transcriptome and metabolome

Based on the enrichment analysis of differential metabolites and individual omics of differential genes in the positive and negative ion mode (Figs. 11A, 11B), the enrichment pathways are ABC transporters, aminoacyl-tRNA biosynthesis, arachidonic acid metabolism, arginine and proline metabolism, β-alanine metabolism, cysteine and methionine metabolism, galactose metabolism, glutathione metabolism, glycerophospholipid metabolism, histidine metabolism, pyrimidine metabolism, starch and sucrose metabolism, steroid biosynthesis, tricarboxylic acid cycle, ether lipid metabolism, glycine, serine and threonine metabolism, glyoxylate and dicarboxylate metabolism, phenylalanine metabolism, pyruvate metabolism, taurine and hypotaurine metabolism, tyrosine metabolism, valine, and leucine and isoleucine biosynthesis. Although most of the understanding of the biology and function of arachidonic acid metabolites comes from mammalian biology research, these metabolites can also be produced by lower eukaryotes, including yeast and other fungi. The metabolic pathways of arachidonic acid and the functions of these related compounds in fungal and yeast biology are poorly understood (*Ells et al., 2012*).

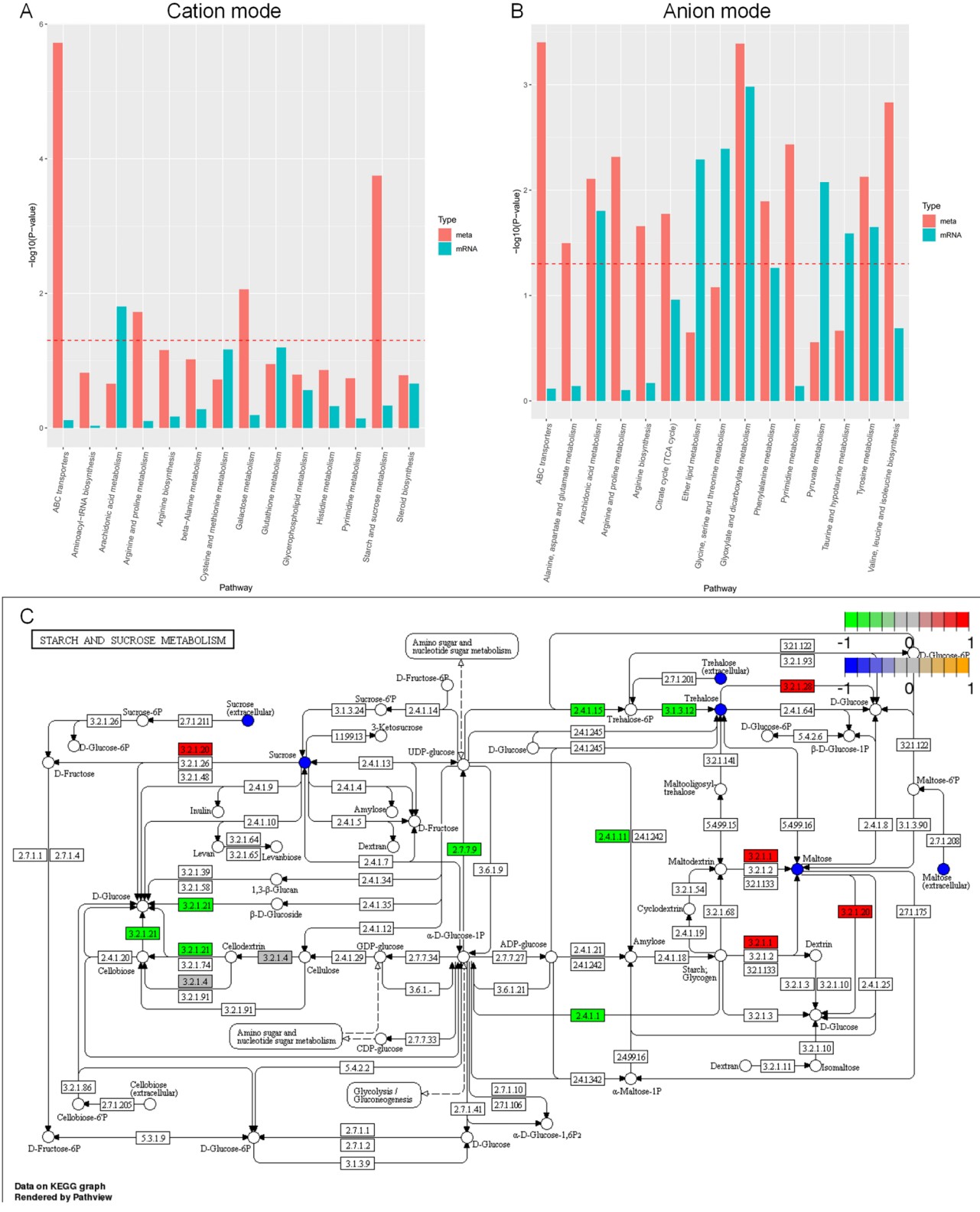

**Figure 11 KEGG enrichment analysis of differential metabolites and individual omics of differential genes.** (A) Bar chart of metabolic pathways by cation mode integrated analysis. (B) Bar chart of metabolic pathways by anion mode integrated analysis. (C) Pathway map of differential metabolites and differential transcripts in starch and sucrose metabolism. The x-axis of the bar chart represents metabolic pathways, the blue bar

**Figure 11 (continued)**
 represents the enriched *p*-value of differential genes, and the red bar represents the enriched *p*-value of differential metabolites, represented by −log (*p*-value). The larger the y-axis, the stronger the significant enrichment degree. Note: α-glucosidase (EC3.2.1.20), α-trehalose glucose hydrolase (EC3.2.1.28), α-amylase (EC3.2.1.1), β-glucosidase (EC3.2.1.21), UDP-glucose pyrophosphorylase (EC2.7.7.9), glycogen phosphorylase (EC2.4.1). 1), glycogen synthetase (EC2.4.1.11), carboxyester hydrolase (EC3.1.3.12).               

Carbon sources play a crucial role in the development of microbial cells. In general, *A. heimuer* has a strong ability to absorb lignin, which involves several regulatory genes. The main enzymes involved in glucose metabolism are glucose glucosidase, glucose hydrolase, glucose pyrophosphorylase, phosphorylase, and glycogen synthetase. However, the regulatory mechanism of *A. heimuer* in the carbon metabolism of crops is still unclear. We mapped the differential metabolites and all expressed transcripts of the D3 *vs*. D6 group to the KEGG pathway database to obtain common pathway information for differential genes ($p < 0.05$) and differential metabolites (VIP > 1, $p < 0.05$). We focused on starch and sucrose metabolism, and the results in Fig. 11C showed that the differential expression levels of metabolites such as sucrose, maltose, and trehalose were relatively high in this pathway. The differential expression levels of α-glucosidase (EC3.2.1.20), α-trehalose glucose hydrolase (EC3.2.1.28), and α-amylase (EC3.2.1.1) are high, as well as the differential expression of β-glucosidase (EC3.2.1.21), UDP glucose pyrophosphorylate (EC2.7.7.9), glycogen phosphorylase (EC2.4.1.1), glycogen synthase (EC2.4.1.11), and carboxyester hydrolase (EC3.1.3.12) transcripts. The main functions of these molecules are catalytic activity and transport activity.

## DISCUSSION

Studies have shown that during the nutritional growth stage, the mycelium of *A. heimuer* mainly utilizes cellulose in the culture medium (*Ma et al., 2022*; *Shan et al., 2024*). From the perspectives of carbon, nitrogen, lignin content, and mycelial growth status, D1 rice straw and D3 corn stalk show relatively good adaptability. From the differences in cellulase activity, the activities of cellulase in D1 and D3 are relatively high, so these two substrates are more suitable for cultivating *A. heimuer* WZD22 and are potential alternative substrates for *A. heimuer*. According to the differences in hemicellulase and ligninase activities, hemicellulase has no obvious correlation with each substrate, and the LACC activity of D3 and D6 is more similar; these results are consistent with previous studies. The extracellular enzymes of *A. heimuer* strains, such as LACC, carboxymethyl cellulase, and filter paper cellulase, have high activities and fast mycelial growth speed (*Li et al., 2024*), which is positively correlated with mycelial growth speed (*Fang et al., 2024*; *Shan et al., 2024*). The extracellular enzymes of *A. heimuer* mycelium show significant differences among most strains. The activities of LACC and cellulase are positively correlated with mycelial growth speed, while xylanase has no obvious correlation (*He et al., 2023*).

From the perspective of correlation analysis, the carbon content and lignin content in the raw materials showed a significant positive correlation, which not only confirmed the

role of lignin as the main carbon source in substrates but also explained why *A. heimuer* mycelium could efficiently utilize lignin-rich substrates such as sawdust. In addition, we also found a significant positive correlation between FPA activity and endoglucanase activity, which further emphasizes the central role of endoglucanase in cellulose degradation and initiating cellulase hydrolysis. It is worth noting that the increase in MnP activity significantly promotes the growth rate of *A. heimuer* hyphae, especially in corn stalks with high lignin content. This discovery not only confirms the potential promoting effect of lignin on mycelial growth (*Song et al., 2018*), but also provides new ideas for optimizing cultivation conditions and increasing yield.

In conclusion, we believe that corn stalks are the best alternative substrate as they provide the conditions necessary for the growth of *A. heimuer* (*Li & Li, 2017*; *Liu et al., 2016*; *Sun et al., 2022*). However, some studies have found that corn cobs are better (*Pan et al., 2016a*; *Sha et al., 2012*), while others indicated that substrates such as soybean stalks and rice straw are also feasible for *A. heimuer* production (*Pan et al., 2016b*; *Wang et al., 2015*; *Zhao, Wang & Liu, 2010*). These phenomena may be related to varietal differences, as the degradation capabilities of *A. heimuer* vary (*He et al., 2023*; *Liu & Song, 2018*), as well as the composition of straw. For the same raw material, the utilization efficiency for different varieties is different, and for the same variety, the utilization of different raw materials also varies.

From the perspective of gene expression profiles, when *A. heimuer* mycelia utilize corn stalks, there exists a complex regulatory mechanism for carbohydrate metabolism. Through metabolic analysis, it is found that the pathways enriched in signal transduction, carbohydrate metabolism and transport, as well as catabolism, are important metabolic pathways for the production of *A. heimuer* using corn stalks (*Shan et al., 2024*). Studying the transcription factors corresponding to these metabolites is conducive to targeted breeding.

## CONCLUSION

This study confirmed that a wild *A. heimuer* mycelium can grow on five different crop substrates and sawdust, but there are significant differences in mycelial growth rate, biomass, and extracellular enzyme activity on different single substrates. According to the growth rate, growth potential, and extracellular enzyme activity, the results showed that *A. heimuer* had the best utilization effect on corn stalk, followed by corn cob, and then rice husk, rice straw, and soybean stalk were the least suitable. The changes of cellulose, hemicellulose, and lignin in *A. heimuer* mycelia cultured with different substrates were investigated. The results showed that *A. heimuer* mycelia had similar decomposition ability to corn stalk and sawdust substrate, and corn stalk may effectively replace sawdust.

Through transcriptome analysis, the top 20 pathways with the highest degree of DEG enrichment were identified among the three comparison groups of rice straw, corn stover, and sawdust substrates. Among them, we focused on significantly enriched redox processes and carbohydrate metabolism processes, including glyoxylate and dicarboxylate metabolism; glycine, serine, and threonine metabolism; glycolysis/gluconeogenesis;

pyruvate metabolism; oxidative phosphorylation; protein processing in endoplasmic reticulum; and ribosome. Starch and sucrose metabolism and carbon metabolism pathways were enriched by KEGG of non-targeted metabolites. By combining multiple factors for statistical analysis, we can better study the effects of different carbon sources on the growth of *A. heimuer* hyphae and differences in metabolites. Therefore, this study not only identified suitable alternative substrates to improve the rational utilization of crop residues but also provided a solid theoretical basis for replacing sawdust substrates when planting *A. heimuer*, contributing to sustainable resource recycling. In the future, our team will conduct research on the production process of corn stalk in order to provide a basis for promoting its application in the cultivation of *A. heimuer*.

### Funding

This project is funded by the Natural Science Foundation of Heilongjiang Province of the People's Republic of China (Project Number: SS2024C003). This research is supported by the Special Fund for Basic Research of Universities in Heilongjiang Province, a scientific research project, with the project number: 2021-KYYWF-0172. The funders had no role in study design, data collection and analysis, decision to publish, or preparation of the manuscript.

### Grant Disclosures

The following grant information was disclosed by the authors:
Natural Science Foundation of Heilongjiang Province of the People's Republic of China: SS2024C003.
Special Fund for Basic Research of Universities in Heilongjiang Province: 2021-KYYWF-0172.

### Competing Interests

The authors declare that they have no competing interests.

### Author Contributions

- Di Zhang conceived and designed the experiments, performed the experiments, authored or reviewed drafts of the article, and approved the final draft.
- Yuchen Liu performed the experiments, analyzed the data, authored or reviewed drafts of the article, and approved the final draft.
- Ying Li analyzed the data, prepared figures and/or tables, and approved the final draft.
- Guosheng Jiang performed the experiments, prepared figures and/or tables, and approved the final draft.
- Mingzhu Meng analyzed the data, authored or reviewed drafts of the article, and approved the final draft.
- Jihua Wang conceived and designed the experiments, prepared figures and/or tables, authored or reviewed drafts of the article, and approved the final draft.

## Data Availability

The raw data is available in the Supplemental File.

## Supplemental Information

Supplemental information for this article can be found online at http://dx.doi.org/10.7717/peerj.19300#supplemental-information.

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
