# Peer review of "Evaluation and transcriptomic and metabolomic analysis of the ability of Auricularia heimuer to utilize crop straw"

_PeerJ, doi:10.7717/peerj.19300_

## Round 0.1 · original submission · Major Revisions

In addition to the points raised by the reviewers, could you please discuss your results in the context of two recent papers (if you think they are relevant):

Transcriptome analysis reveals key genes involved in the degradation of corn cobs by Auricularia heimuer
Nannan Geng … Li Zou
Industrial Crops and Products, Volume 224, February 2025, 120417; https://www.sciencedirect.com/science/article/pii/S092666902402394X

A Combination of Transcriptome and Enzyme Activity Analysis Unveils Key Genes and Patterns of Corncob Lignocellulose Degradation by Auricularia heimuer under Cultivation Conditions
Ming Fang … Evans Kaimoyo
J. Fungi 2024, 10(8), 545; https://doi.org/10.3390/jof10080545

·

Basic reporting

It is recommended to change the "Results" section into "Results and Discussion".

Experimental design

In the "Preparation of Culture Medium" section of the Materials and Methods, the variable is controlled by adjusting the carbon-nitrogen ratio. It is recommended to add references for this method.

Validity of the findings

The result of "Carbon-Nitrogen Ratio of Different Culture Media" is that "The crop substrates most suitable for cultivating the mycelium of Auricularia auricula are rice straw (D1), corn stalk (D3) and soybean stalk (D5)." However, the result of "The Influence of Different Substrates on Mycelium Growth" is that "The mycelium of D5 is relatively sparse. And D4 has a good effect on cultivating the mycelium of Lentinula edodes." Are these two results contradictory? It is recommended to discuss this in the text.

·

Basic reporting

1. Clarify the enzyme activity represented by the vertical axis in Figure 3.
2. Standardize all expression to either U/mL or mL.
3. Figures 7 and 11 are blurry. Please provide images with at least 300 ppi resolution, as required by the journal.
4. Although the introduction mentions that the shortage of raw materials and the rising cost of wood chips, the traditional substrate for Auricularia heimuer cultivation, has led to the exploration of agricultural straw as an alternative substrate, what are the current limitations and potential impacts of this alternative? Please provide further elaboration.
5. Regarding the results of elemental analyses, in addition to reporting the carbon and nitrogen contents and C/N ratios of the various substrates, how do these data influence the growth of Auricularia heimuer? Please expand on this analysis.
6. What is the efficiency of nutrient uptake and utilization by these mycelia across different substrates? The authors currently only describe the effects of various substrates on mycelial growth through intuitive measures such as growth rate, density, and color.
7. The results regarding enzyme activity lack a comprehensive discussion. Please include a comparison of the enzyme activity of other similar edible and medicinal fungi (EMFs) grown on the same or similar substrates.
8. Which mathematical model was used for the correlation analysis: Pearson's or Spearman's correlation, or another model?
9. The correlation analysis section still lacks an in-depth discussion. While the correlation between the carbon content of the substrate, enzyme activity, and other factors with the growth of Auricularia heimuer mycelium is acknowledged, further exploration is needed to investigate potential interactions between these factors and how they collectively influence the growth of Auricularia heimuer. Please focus on adding this analysis.
10. In the transcriptome analysis results, in addition to the functional annotation and enrichment analysis of differentially expressed genes, could their specific mechanisms in regulating the strain's adaptation to different substrates be discussed in more detail, in conjunction with the results of your investigation?
11. Could KEGG enrichment provide further insights or explanations regarding the role of these differential metabolites in the growth process of Auricularia heimuer?

Experimental design

Refine the method for determining cellulase, hemicellulase, and ligninase activities so that other researchers can more easily replicate the experiment.

Validity of the findings

no comments

Additional comments

Please carefully format your references according to the formatting requirements of the PeerJ, many articles are missing page numbers.

·

Basic reporting

After reviewing the article titled "Evaluation and transcriptomic and metabolomic analysis of the ability of Auricularia heimuer to utilize crop straw," I have the following specific suggestions for the authors to consider and improve upon. The authors have done a lot, but the writing requires significant improvement:

When introducing the topic of crop straw, it is recommended that the article analyzes from a global perspective, as crop straw has always been a hot issue worldwide.
Lines 48-80 could be divided into two paragraphs, one explaining the importance of crop straw and the other introducing the relevant enzymes.

Experimental design

In the experimental design section, it is suggested that the authors provide more specific details about the experimental operations, such as the screening and purification process of the strain W-ZD22, as well as detailed information on culture conditions (such as temperature, humidity, light, etc.), so that other researchers can replicate the experiment.
Regarding the correlation heatmap in Figure 5, the authors are advised to provide more explanation and discussion on how the correlation between different substrates, enzyme activities, and mycelial growth rates affects the cultivation efficiency of Auricularia heimuer.
Why is the color inconsistent in Figures 3 and 4?
The annotations for some figures (such as Figure 5) are not detailed enough; it is recommended to add necessary labels and explanations to help readers better understand the content of the figures.

Validity of the findings

Line 195, is the results section merged with the discussion section? Please refer to the journal's format to determine the writing style for the results and discussion sections. Additionally, the article indeed discusses this section. There is no clear separation between the discussion and results sections, and this part is very confusing.
In the discussion section, many parts do not have appropriate references; please insert relevant literature in the appropriate places.
The discussion section could be further expanded to include comparisons of the effects of different culture substrates used in other studies on the growth of Auricularia heimuer, as well as the similarities and differences between this study's results and those studies.
The conclusion section needs to more accurately reflect the research results, especially regarding the potential and practical prospects of using corn straw as an alternative culture substrate.
At the end of the article, it is suggested that the authors propose future research directions based on the current research results, such as exploring the utilization efficiency of agricultural waste by Auricularia heimuer under different environmental conditions or studying its potential application in actual agricultural production.

---

## Round 0.2 · Minor Revisions

Please respond to the remaining questions and comments of the third reviewer. If you do not agree with them, please explain.

·

Basic reporting

no comment

Experimental design

no comment

Validity of the findings

no comment

Additional comments

no comment

·

Basic reporting

The author has made all the necessary modifications as requested. Therefore, I suggest accepting it.

Experimental design

no comment

Validity of the findings

no comment

Additional comments

no comment

·

Basic reporting

it ok now

Experimental design

it good now

Validity of the findings

Why did the author not reply to these questions?
Validity of the findings

Line 195, is the results section merged with the discussion section? Please refer to the journal's format to determine the writing style for the results and discussion sections. Additionally, the article indeed discusses this section. There is no clear separation between the discussion and results sections, and this part is very confusing.

In the discussion section, many parts do not have appropriate references; please insert relevant literature in the appropriate places.

The discussion section could be further expanded to include comparisons of the effects of different culture substrates used in other studies on the growth of Auricularia heimuer, as well as the similarities and differences between this study's results and those studies.

The conclusion section needs to more accurately reflect the research results, especially regarding the potential and practical prospects of using corn straw as an alternative culture substrate.

At the end of the article, it is suggested that the authors propose future research directions based on the current research results, such as exploring the utilization efficiency of agricultural waste by Auricularia heimuer under different environmental conditions or studying its potential application in actual agricultural production.

---

## Round 0.3 · Minor Revisions

The substance is fine, but there remain misprints, e.g.:
line 28: "differentexpression";
line 104: "These transcriptomic information;
names in Discussio set in capital letters.
{lease proofread the text carefully.

---

## Round 0.4 · accepted · Accept

Some language problems remain, e.g. "different expression genes (DEGs)" should be "differentially expressed genes". I will accept the paper based on the scientific evaluation, but I suggest to proofread it carefully.